# *CHI3L1*, *NTRK2*, *1p/19q* and *IDH* Status Predicts Prognosis in Glioma

**DOI:** 10.3390/cancers11040544

**Published:** 2019-04-15

**Authors:** Elise Deluche, Barbara Bessette, Stephanie Durand, François Caire, Valérie Rigau, Sandrine Robert, Alain Chaunavel, Lionel Forestier, François Labrousse, Marie-Odile Jauberteau, Karine Durand, Fabrice Lalloué

**Affiliations:** 1EA3842 CAPTuR, Faculty of Medicine, University of Limoges, 2 Rue du Docteur Marcland, 87025 Limoges, France; elise.deluche@unilim.fr (E.D.); barbara.bessette@unilim.fr (B.B.); sandrine.robert@unilim.fr (S.R.); Alain.Chaunavel@chu-limoges.fr (A.C.); francois.labrousse@unilim.fr (F.L.); m-o.jauberteau-marchan@unilim.fr (M.-O.J.); karine.durand@chu-limoges.fr (K.D.); 2Department of Medical Oncology, Limoges University Hospital, 2 rue Martin Luther King, 87042 Limoges, France; 3Bioinformatics Team, BISCEM Platform, CBRS, University of Limoges, 2 rue du Docteur Marcland, 87025 Limoges, France; stephanie.durand@unilim.fr (S.D.); lionel.forestier@unilim.fr (L.F.); 4EA7500 PEREINE, University of Limoges, 123 av. Albert Thomas, 87060 Limoges, France; 5Department of Neurosurgery, Limoges University Hospital, 2 rue Martin Luther King, 87042 Limoges, France; francois.caire@chu-limoges.fr; 6Department of Neuropathology and INSERM U1051, Hospital Saint Eloi—Gui de Chauliac, 80 av. Augustin Fliche, 34090 Montpellier, France; v-rigau@chu-montpellier.fr; 7Department of Pathology, Limoges University Hospital, 2 rue Martin Luther King, 87042 Limoges, France; 8Department of Immunology, Limoges University Hospital, 2 rue Martin Luther King, 87042 Limoges, France

**Keywords:** glioma, *IDH* status, *CHI3L1*, *NTRK2*, prognosis

## Abstract

The aim of this study was to identify relevant biomarkers for the prognosis of glioma considering current molecular changes such as *IDH* mutation and 1p19q deletion. Gene expression profiling was performed using the TaqMan Low Density Array and hierarchical clustering using 96 selected genes in 64 patients with newly diagnosed glioma. The expression dataset was validated on a large independent cohort from The Cancer Genome Atlas (TCGA) database. A differential expression panel of 26 genes discriminated two prognostic groups regardless of grade and molecular groups of tumors: Patients having a poor prognosis with a median overall survival (OS) of 23.0 ± 9.6 months (group A) and patients having a good prognosis with a median OS of 115.0 ± 6.6 months (group B) (*p* = 0.007). Hierarchical clustering of the glioma TCGA cohort supported the prognostic value of these 26 genes (*p* < 0.0001). Among these genes, *CHI3L1* and *NTRK2* were identified as factors that can be associated with *IDH* status and 1p/19q co-deletion to distinguish between prognostic groups of glioma from the TCGA cohort. Therefore, *CHI3L1* associated with *NTRK2* seemed to be able to provide new information on glioma prognosis.

## 1. Introduction

Gliomas are the most common primary brain tumors of the central nervous system. Overall age-adjusted incidence rates vary from 4.67 to 5.73 per 100,000 persons [1]. Gliomas are classified according to World Health Organization (WHO) criteria [2]. However, despite the undeniable contribution of this classification, patients with tumors that have the same histologic appearance may still have different outcomes due to molecular heterogeneity [3].

One way to improve the clinical management of gliomas is to identify new molecular biomarkers that distinguish more homogeneous subgroups of patients regardless of histological tumor characteristics, which could further refine the prognostic value of the current biomarkers, 1p19q chromosome arm loss and *IDH* mutation. These biomarkers could also improve our knowledge of glioma biology. 

*IDH* mutations are strongly implicated in tumor initiation and progression [4], and the presence or absence of *IDH* mutations contributes to glioma prognosis [5]. However, while *IDH* mutations are a hallmark of diffuse low-grade glioma (LGG), as they occur in 70 to 90% of astrocytomas and oligodendrogliomas (WHO grades II and III), they are also identified in 10% of glioblastomas (GBMs) (WHO grade IV) [2], which are associated with a better prognosis as compared to *IDH* wild-type GBM [6,7,8]. Despite the prognostic benefit associated with *IDH* mutations, and regardless of the glioma subtype, differences in outcomes among *IDH*-mutant tumors have been observed [9]. Deletion of 1p19q is a characteristic of oligodendrogliomas, where it occurs concomitantly with *IDH* mutation. It is a strong independent prognostic biomarker associated with improved survival [10]. This glioma group, although relatively homogeneous on the molecular level, could also have different outcomes [11]. 

Efforts have been focused on classifying gliomas according to molecular aberrations and resultant new genetic signatures that could help to optimize the clinical management [12,13]. To improve prognosis prediction, other genes associated with *IDH* and 1p19q status need to be identified. So, in this study we analyzed 96 genes considered as relevant based on our previous work [14]. These potential candidate prognostic biomarkers are usually conventional genes involved in different mechanisms implicated in gliomagenesis, such as signaling pathways, hypoxia, angiogenesis, or cancer stem cell markers. More rarely, these genes are involved in the glycosylation process or in neurotrophin receptor-dependent signaling. Aberrant glycosylation is a common feature of different tumor types and is generally associated with deregulation of cell adhesion or migration, which results in tumor processes such as metastasis and invasion. Our previous study allowed us to present evidence that eight glycosylation-related genes were upregulated in the most aggressive and undifferentiated glioblastoma cells [14]. In parallel, the family of tyrosine receptor kinases (TRKs) and p75NTR are already known to be involved in tumor cell survival. For instance, TrkB and TrkC receptors promote the growth of brain tumor-initiating cells, and p75NTR promotes glioma invasion [15,16,17]. Finally, we recently showed that some autophagic factors and neurotrophin pathways cooperate to contribute to tumor cell aggressiveness [18]. Thus, genes encoding Trks and glycosylation-related genes could be of interest to complement the current IDH mutation-based classification.

In this study, we screened expression levels of a set of genes to identify those linked to prognosis and analyzed whether they could refine prognosis according to *IDH* and 1p19q status in glioma. 

## 2. Results 

### 2.1. Clinical and Histological Characteristics

The analysis was performed in radiotherapy- and/or chemotherapy-naive glioma surgical specimens. Sixty-four patients (26 women and 38 men; median age: 47.5 years; range: 22–81) were included. 

The main clinical and histological data from this treatment-naive population are shown in Appendix A. The population comprised patients with gliomas ranging from grade II to grade IV (grade II, *n* = 18; grade III, *n* = 25; grade IV, *n* = 21). Thirty-six tumors exhibited the *IDH* mutation and 28 were *IDH* wild-type; 14 tumors had 1p19q deletion. 

### 2.2. Hierarchical Clustering of Studied Cohort to Define Two Distinct Prognostic Groups

Using a 96-gene set with hierarchical clustering identification, two distinct groups were identified (data not shown). To better identify differentially expressed genes between the two groups, univariate analysis and multivariate analysis (principal component analysis) were performed. Univariate analysis detected 29 genes of interest (*p* < 0.05) (Appendix A, left), whereas principal component analysis detected 30 genes of interest (Appendix A, right). By combining these two techniques, we identified a relevant 26-gene panel (Appendix A): Nine genes involved in neurotrophin pathways (*AKT3*, *EGF*, *AREG*, *ERBB4*, *neuregulin 2*, *neuregulin 3*, *neuregulin 4*, *BRAF*, *internexine-a*), six genes encoding neurotrophins and their receptors (*NGF*, *NTSR2*, *NTRK1*, *NTRK2-FL*, *NTRK2-T1*, *NTRK3*), three glycosylation-related genes (*CHI3L1*, *KLRC3*, *ST3GAL5*), two genes implicated in autophagy (*PARK2*, *PINK1*), three genes known to occur in hypoxia and angiogenesis (*VEGF-A*, *VEGFR-2*, *VEGFR-3*) and three glioma markers (*OLIG2*, *NANOG*, *SYP*).

Using this final 26-gene set, new hierarchical clustering classified the 64 tumors into two prognostic groups, A and B (Figure 1a), which were highly discriminative in predicting prognosis. Group A identified patients having a poor prognosis with a median survival of 23.0 ± 9.6 months, whereas group B identified patients having a good prognosis with a median survival of 115.0 ± 6.6 months (*p* = 0.007) (Figure 1b).

Groups A and B showed distinct clinical and histological characteristics (*p* < 0.05). Group A mainly contained astrocytomas, including glioblastoma and *IDH* wild-type tumors; patients were older (*p* = 0.0024) and received more radiotherapy than those in group B (*p* = 0.04). Group B contained more grade II and *IDH-*mutated tumors than group A. There was no difference between the two groups in tumor localization, surgery, chemotherapy, and 1p19q status (*p* > 0.05) (Appendix A).

### 2.3. Validation of Clustering in the Independent The Cancer Genome Atlas (TCGA) Cohort

To confirm the relevance and ability of groups A and B to discriminate between patient outcomes, we analyzed the 26 genes of our molecular signature by principal component analysis in an independent glioma TCGA cohort that included 671 tumors: Only primary glioblastoma (de novo, untreated; *n* = 157), grade II (*n* = 249) and grade III (*n* = 265) gliomas.

Hierarchical clustering in a TCGA cohort dedicated to gliomas clearly distinguished two tumor groups (Figure 2a). These results confirmed those we obtained with our cohort, as patients in group A had a poor prognosis with a median survival of 16.8 ± 2.0 months, whereas patients in group B had a good prognosis with a median survival of 105.0 ± 17.3 months (*p* < 0.0001) (Figure 2b).

Our cohort and the TCGA cohort were quite similar concerning the main histomolecular characteristics, including age (median age of 46 years; range: 18–89), tumor grade, and histology (Appendix A).

Within the 26 genes that were differentially expressed in groups A and B, the expression levels (defined by median values) were significantly or nearly significantly associated with prognosis in both cohorts; only *NGF* lost its prognostic significance in the TCGA cohort.

### 2.4. TCGA Cohort and Definition of Prognosis Groups

We found that the prognosis values determined by tumor grade and current 1p19q/*IDH* status could be further refined when stratified on our group signature (Figure 3).

For example, while median overall survival (OS) of patients with grade II tumors was 117 months (Figure 3a), stratification of groups A and B predicted a median OS of 62 months vs. 130 months for grade II/group A and grade II/group B, respectively (Figure 3b).

The same observation could be made for patient prognosis with grade III and IV tumors alone (Figure 3a) and further stratified on groups A and B (Figure 3b).

To the same extent, current *IDH* and 1p19q status provided additional prognostic information when further stratified on groups A and B (Figure 3c,d). For example, median OS of patients with *IDH* wild-type was 15 months, whereas the stratification on group A and B predicted a different prognosis (117 months). The same observation was found for patients having *IDH* mutation with or without 1p19 co-deletion (Figure 3c,d).

### 2.5. Identification of Two Relevant Independent Prognostic Biomarkers

Among the 26 selected genes, we identified two genes that appeared to be most relevant because they were significantly differentially expressed between the two groups when defined by median threshold and might influence prognosis: *NTRK2* (median: 14.76) and *CHI3L1* (median: 8.98). The expression levels of these genes were subsequently analyzed independently. In our cohort (Figure 4a,c) and the glioma TCGA cohort (Figure 4b,d), low expression of *NTRK2* and high expression of *CHI3L1* were strongly linked to poor prognosis (*p* < 0.05).

Combined molecular groups based on the presence or absence of the four tumor markers (*IDH* status, *1p19q* co-deletion status and *CHI3L1* and *NTRK2* expression levels) were used to classify the 671 glioma cases in the TCGA cohort according to prognosis. On their own, *IDH* and *1p19q* co-deletion status determined three outcomes: Poor, intermediate and good, with median OS of 15, 80 and 134 months, respectively (*p* > 0.0001, Figure 5a). Adding *CHI3L1* and *NTRK2* to that prognosis panel, 12 subsets were identified (Figure 5a). Then three molecular groups were identified by grouping similar prognostic populations considering all the parameters (Figure 5a,b): (1) Better prognosis group with 1p19q co-deletion, *IDH* mutation, low *CHI3L1* expression and high *NTRK2* expression; (2) poorer prognosis group with *IDH* wild-type and high *CHI3L1* expression regardless of *NTRK2* expression; and (3) intermediate prognostic group with the other combinations (Figure 5b). Prognostic populations were found in each grade of glioma, suggesting that the prognostic scheme could be applied independently in grade II, grade III and grade IV gliomas (Figure 5c).

These molecular combined groups identified a better prognostic group among *IDH*-mutated and 1p19q co-deleted tumors based on *CHI3L1* and *NTRK2* expression. Median survival in the group with *IDH* mutated, 1p19q co-deleted, low *CHI3L1* expression and high *NTRK2* expression (*n* = 81) was significantly higher than in the group with *IDH* mutated, 1p19q co-deleted and other association of *CHI3L1* and *NTRK2* expression (*n* = 105) (154 months vs. 75 months; *p* = 0.0001). Similarly, a poorer prognosis group among *IDH* wild-type tumors based on *CHI3L1* expression was highlighted. The median survival in the group with *IDH* wild-type and high *CHI3L1* expression (*n* = 235) was 14 months compared to 117 months in the group with *IDH* wild-type and low *CHI3L1* (*n* = 15).

In addition, multivariate logistic regression analysis demonstrated that low *CHI3L1*, high *NTRK2*, grade and 1p19q co-deletion seemed to be independent predictors for *IDH* mutation (*p* ≤ 0.0001) (Table 1).

Multivariate Cox model analysis revealed independent markers of prognosis: Age of diagnosis (hazard ratio (HR): 1.039; 95% confidence interval (CI): 1.028–1.051; *p* < 0.0001), grade of glioma (*p* < 0.05) and molecular combined group (*p* < 0.05) (Appendix A).

*CHI3L1* and *NTRK2* could be added to *IDH* status and 1p19 co-deletion to further refine prognosis, as summarized in Figure 6.

## 3. Discussion

In this study, we analyzed a 96-gene set to improve prognostic prediction in glioma, notably when based on *IDH* and 1p19q status. Since radiotherapy can modify gene expression [19], we only analyzed the glioma samples resected from patients who had not undergone radiotherapy and/or chemotherapy. A 26-gene signature identified two molecular subgroups that were significantly correlated with patient survival and validated on an independent TCGA cohort.

One way to refine the management of glioma is to improve prognosis by examining molecular information, which can be more precise than histology alone [20,21]. Previous studies evaluated the impact of molecular classification on glioma prognosis. Moreover, some studies showed that gene expression profiling provides additional information to discriminate between glioma subtypes from grade II to IV [22,23,24]. The present study identified specific genes related to glycosylation and genes coding for neurotrophins and their receptors in the same molecular signature. We examined how they could improve (or not) prediction of glioma prognosis, in particular when associated with 1p19q loss and *IDH* mutational status. For the first time, we propose that *CHI3L1* and *NTRK2* could act as new biomarkers to improve the assessment of glioma prognosis.

*NTRK2* encodes neurotrophin receptors, which are well known to play a role in brain tumor pathogenesis [25]. In the present study, the *NTRK2* level was significantly higher in grade II than in grade IV tumors (*p* < 0.05). Moreover, in the grade III and IV subgroups, the survival rate was worse with low *NTRK2* expression than with high expression [26]. These results may be related to the role of *NTRK2* in invasiveness and gliomagenesis in early astrocytoma [27,28,29]. In agreement with that, we showed in previous studies that TrkB and p75^NTR^ were expressed in glioblastoma cell lines and could be required for tumor aggressiveness [15,16].

*CHI3L1*, a gene encoding YKL-40, a marker of the glioblastoma mesenchymal subtype, is overexpressed in glioblastoma cancer stem cells [30]. Inactivation induces a loss of aggressiveness and leads to modification of neurotrophin receptor expression [15]. *CHI3L1* is more highly expressed in glioblastoma than in normal brain [31,32]. Previously, we observed in glioblastoma cell lines that *CHI3L1* might be a cancer stem cell biomarker in glioma, based on glycosylation-related gene expression analysis, and may be a potential biomarker of aggressiveness [14]. Thus, the present study confirmed that *CHI3L1* expression significantly differed between grade II and III glioblastoma and that changes in *CHI3L1* expression level were associated with glioma prognosis, as reported by Steponaitis et al. [33]. A recent meta-analysis of 1241 glioblastoma patients confirmed our results by showing that high *CHI3L1* expression was associated with poor prognosis (HR  =  1.46; 95% CI, 1.33–1.61; *p*  <  0.001) [31].

Since a recent study demonstrated that CHI3L1 is less expressed in IDH-mutated glioblastoma [34], we determined whether IDH status was connected with other independent molecular factors by multivariate analysis. The relationship between IDH status and NTRK2 [35] or CHI3L1 [36] has not yet been studied in grade II and III tumors and was only reported for glioblastoma.

Molecular analysis previously established that it was possible to define patient groups based on *IDH* status and 1p19q co-deletion status, as described previously [6,37]. In this study we identified different prognostic subgroups based on *CHI3L1* and *NTRK2* expression levels associated with *IDH* status and 1p19q co-deletion status. *IDH* mutation, 1p19q co-deletion, low *CHI3L1* expression and high *NTRK2* expression characterized patients with a better prognosis, whereas *IDH* wild-type, absence of 1p19q co-deletion and high *CHI3L1* expression defined patients with a poor prognosis.

Our data are in agreement with those showing that *IDH* wild-type glioblastoma with overexpression of NTRK2 is associated with better OS (*p*  =  0.049; HR: 0.66) [35]. Furthermore, NTRK2 was associated with peroxisome proliferator–activated receptor α (PPARα), which was overexpressed and correlated with a good prognosis in *IDH* wild-type primary glioblastoma [38]. *CHI3L1* overexpression was associated with mesenchymal subtype in glioblastoma (defined by *IDH* wild-type) [36].

Indeed, introducing the *IDH* mutation into primary human astrocytes alters specific histone markers and induces extensive DNA hypermethylation in a manner consistent with glioma-CpG island methylator phenotype (G-CIMP), which is associated with better clinical outcomes for patients, as described for glioma [39]. In addition, G-CIMP–positive tumors may be less aggressive due to silencing of key mesenchymal genes in glioblastoma [40]. Since promoter methylation of *CHI3L1* leads to the loss of mesenchymal properties [41], it has been suggested that a relationship may exist between *CHI3L1* expression and *IDH* mutation by inducing *CHI3L1* promoter methylation [39]. The hypermethylation phenotype of *CHI3L1* could also result in low *CHI3L1* expression and good prognosis. However, in this study, some patients with *IDH* mutations had different gene profiles with high *CHI3L1* expression and an intermediate prognosis. In this case, variations in *CHI3L1* expression may reflect another methylation mechanism involving additional pathways, such as those comprising phospho-c-jun and DNMT1 [41]. Unlike *CHI3L1* expression, the role of methylation in the *NTRK* family has not been demonstrated in glioma [26].

Based on the data presented here, we suggest that *CHI3L1* and *NTRK2* are two potential surrogate markers to identify certain subgroups with distinct prognosis. The findings are not a revision of classification strategies, but rather a window into potential targeted treatments. Subsequently, CHI3L1-targeted therapy could be used to treat patients/tumors with molecular profiles associated with a poor prognosis (*IDH* wild-type, 1p19 co-deletion absent, high *CHI3L1* and low *NTRK2*). It has already been reported that CHI3L1 targeting affords an effective treatment for glioma in animal models [42,43].

In the future, better treatment outcomes for patients with a poor prognosis (*IDH* wild-type, absence of 1p19 co-deletion, high *CHI3L1* expression and low *NTRK2* expression) might be proposed by combining conventional chemotherapy (radiotherapy and temozolomide) with CHI3L1-targeted inhibitors, especially in grade IV [44]. In grade II patients with a very good prognosis, therapeutic de-escalation or treatment with NTRK2 inhibitor may be considered.

These results will have to be validated in a prospective investigation in order to develop a robust protocol for microarray analysis, to test the interobserver reproducibility of our approach and to confirm the prognostic value of these biomarkers and/or their potential as therapeutic targets. *CHI3L1* and *NTRK2* status could be investigated in tumors by using frozen tissues, or in the future by using formalin-fixed paraffin-embedded sections to facilitate their use in routine practice [45].

## 4. Materials and Methods

### 4.1. Patients and Tumor Samples

Inclusion criteria were a minimum age of 18 years and availability of clinical and survival data. Glioma samples were obtained from 64 adult patients who underwent surgery at Limoges and Montpellier University Hospitals from 1993 to 2013 and did not have any treatment such as chemotherapy and radiotherapy. Clinical and survival data were obtained by a retrospective query, and all samples were used in accordance with French bioethics laws regarding patient information and consent.

This study benefited from the expertise of Prof. Labrousse (collection manager of Limoges), Prof. Rigau (collection manager of Montpellier) and Prof. Sylvain Lehmann (manager of the Biological Resource Center of Montpellier University Hospital (CRB-CHUM)). Gliomas were classified according to the 2016 WHO classification.

All samples were visually inspected at the time of this study for their tumor content. Histologic diagnoses were made on formalin-fixed, paraffin-embedded sections and tumor sections were stained with hematoxylin phloxine saffron. Paraffin-embedded blocks were obtained from the Tumor Biobank (CRBiolim; http://www.crbiolim.fr/) of Limoges University Hospital, following research project approval by the Institutional Review Board (AC-2013-1853, DC-2011-1264), and from CRB-CHUM (http://www.chu-montpellier.fr; Biobank ID: BB-0033-00031).

The routine molecular diagnosis workflow used tumoral DNA from formalin-fixed paraffin-embedded (FFPE) sections or, when genomic DNA was too degraded as estimated by our in-house qPCR qualification method, tumoral DNA from snap-frozen tissue.

Control brain tissues (2 samples) were collected from the cerebral parenchyma of a breast cancer metastasis in a 50-year-old woman. Only the tissue located farthest from the metastasis was retained, and histological analysis was carried out to confirm the absence of tumor tissue.

### 4.2. Detection of IDH Mutations by Pyrosequencing

Tumoral genomic DNA (10 ng) was amplified in a 25 μL PCR reaction using the Pyromark PCR kit (Qiagen, Hilden, Germany); PCR and sequencing primers were obtained from Wang et al.’s published method [46]. All experiments were performed as described by the authors.

### 4.3. Detection of 1p and 19q Chromosome Arm Loss or Retention by Multiplex-Ligation-Proximity Assay

The multiplex-ligation-proximity assay (MLPA) technique was used to analyze 1p and 19q chromosome arm loss or retention, according to the manufacturer’s recommendations (SALSA^®^ MLPA^®^ P088-C2 Oligodendroglioma kit, MRC-Holland, Amsterdam, The Netherlands). Probe sequences and locations are available on the manufacturer’s site. Briefly, after MLPA experiments, denatured fragments were separated according to their size by capillary electrophoresis on an ABI 3130XL capillary sequencer (Applied Biosystems, Waltham, MA, USA) and peak heights were quantified by GeneMapper software (Applied Biosystems). Data analysis was performed using an in-house Microsoft Excel matrix as previously described [47,48].

### 4.4. Total RNA Extraction from Human Tumor Samples

Before RNA extraction, 5 µm thick frozen tissue sections were histologically reviewed by a neuropathologist to ensure that they contained a minimum of 60% tumor cells. Tumor tissue (4–40 mg) was pulverized with CK14 ceramic balls (Ozyme, Frankreich, Franch) in a lysis buffer (QiaZol Lysis Reagent, Qiagen) using a Precellys 24 homogenizer (Précellys24^®^, Ozyme). RNA was extracted from lysed tissues according to the manufacturer’s protocol (Qiagen). RNA concentration was determined by spectrophotometry (NanoDrop ND1000, Labtech, Vancouver, WA, USA), and integrity was assessed by capillary electrophoresis (RNA 6000 Nano Kit, Bioanalyzer 2100, Agilent Technologies). All samples used in this study had an RNA integrity number (RIN) greater than 6. Complementary DNA (cDNA) was synthesized from 1 μg of total RNA using the High Capacity cDNA Reverse Transcription Kit^®^ (Life Technologies, Carlsbad, CA, USA) according to the manufacturer’s protocol.

### 4.5. TaqMan Low-Density Array

**Design:** The custom-made TaqMan low-density array (TLDA) card contained 4 identical 96-gene sets including glioma markers, genes coding for neurotrophins and their receptors and genes involved in different mechanisms such as glycosylation, autophagy, receptor tyrosine kinase signaling, hypoxia and angiogenesis (Applied Biosystems). The complete list is available in Appendix A.

Each card contained housekeeping genes: *HPRT1*, *18S*, *GAPDH* and *β2-microglobulin*. Interexperiment reproducibility was verified on 4 samples that were analyzed twice, and intraexperiment reproducibility was verified by duplicate amplification of a single gene (*PRUNE2* or *KIAA0367*). The amplification protocol and data analysis were carried out as previously described by Ermonval et al. [49].

**Analysis:** Gene expression profiling: Data analysis was performed using the ΔΔCt method with normalization of the raw data to housekeeping genes [50]. The NormFinder algorithm was used to determine the optimal normalizer gene among the 4 housekeeping genes used. Subsequently, after normalization of the data to *HPRT1*, the mean of pooled samples from normal brain was used to calculate fold changes (tumor/normal ratio). The significance of differences in means between log_2_-transformed fold changes was tested using a 2-tailed Student t-test after confirming homoscedasticity by Fisher test. Significance was set at *p* < 0.05. Exploratory analyses of mRNA expression data were conducted in R (version 3.4.2) (https://www.R-project.org) after log_2_ transformation of fold changes.

The International Society of Neuropathology–Haarlem consensus guidelines for nervous system tumor classification and grading recommends grade evaluation before molecular analysis [51]. Thus, from the expression levels of the 92 genes, a subset of genes able to discriminate between WHO grades was selected on the basis of 2 statistical analyses: (i) Genes showing a significant difference between grade II, III and IV tumors by *t*-test (*p* < 0.05); and (ii) genes highly correlated with grade II, III or IV tumors. Principal component analysis (PCA) was performed with the FactoMineR package [52] and hierarchical clustering (HC) with the ComplexHeatmap package [53].

To distinguish low and high expression of a gene, the median cutoff was chosen as previously described in the literature [54].

### 4.6. Validation of Gene Expression Profiling

The classification of gliomas by PCA and HC based on our expression dataset was validated in an independent cohort of glioblastoma and grade II and III datasets from The Cancer Genome Atlas (TCGA) project. Normalized (RSEM method) and batch-corrected RNAseq expression data corresponding to whole gliomas (named GBMLGG cohort) were downloaded from the Firehose portal of the Broad Institute (gdac.broadinstitute.org). Clinical information, IDH mutation, and 1p19q status were also retrieved from the Broad GDAC portal. All clinical data were downloaded from the TCGA portal. Gliomas were classified according to the 2016 WHO classification.

### 4.7. Statistical Analysis

Data were analyzed using Statview^®^ (SAS Institute Inc., Cary, NC, USA) and R (www.r-project.org). Quantitative results were expressed as means ± SD. Percentages and medians were compared using parametric or nonparametric tests for ordinal variables depending on the size of the group (chi-squared test, Mann–Whitney test, Kruskal–Wallis test, Student *t*-test). Overall survival (OS) was calculated from the date of initial surgery/biopsy to the date of death or last follow-up. Logistic regression was used to analyze the factors associated with *IDH* status. Survival curves were obtained using the Kaplan–Meier method. Relevant variables associated with OS were examined using univariate and, where applicable, multivariate Cox proportional hazards regression. For the multivariate models, a univariate inclusion criterion of *p* ≤ 0.2 was used. Tests or comparisons were considered significant when *p* ≤ 0.05.

## 5. Conclusions

In conclusion, it appears increasingly necessary to coordinate histology results with molecular marker analysis. *CHI3L1* and *NTRK2*, whose expression is associated with *1p19q* co-deletion and *IDH* status, refined the prognosis of glioma patients, but this should be confirmed by a prospective study.

## Figures and Tables

**Figure 1 cancers-11-00544-f001:**
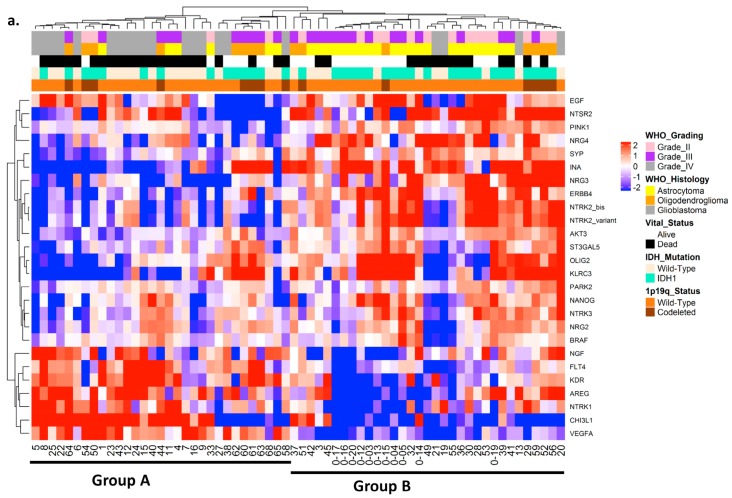
(**a**) Hierarchical clustering of brain tumors based on expression profiles of 26 genes in studied cohort: log2-transformed and mean-centered fold-change values from 26 genes fulfilling the criterion of statistically significant deregulation between two stages of glioma (*p* < 0.05). Red and blue indicate transcript levels above and below median values, respectively. Tumor samples are identified by numbers, and genes by their symbols. Each column indicates the gene expression profile of a sample, and each line indicates variations in the expression level of a given gene among tumor samples. The length of branches on trees forming dendrograms on top of each panel reflects the degree of similarity between samples. Subdivision of samples into two groups according to the dendrogram was subsequently used for survival analysis. (**b**) Overall survival in the studied cohort by Kaplan–Meier analysis: Survival group A (dotted line) defined unfavorable prognosis, and survival group B (solid line) defined favorable prognosis (*p* = 0.007). P-values were calculated using log-rank test. Group A was defined by high expression of *EGF*, *NGF*, *FLT4*, *KDR*, *AREG*, *NTRK1*, *VEFGA* and *CHI3L1* and low expression of *SYP*, *INA*, *ST3GAL5*, *OLIG2*, *KLRC3*, *NTSR2*, *PINK1*, *NRG3*, *ERBB4*, *NTRK2-FL*, *NTRK2-T1*, *AKT3*, *PARK2*, *NANOG*, *NTRK3*, *NRG2*, *BRAF* and *NRG4*, whereas group B was defined by the opposite pattern of gene expression.

**Figure 2 cancers-11-00544-f002:**
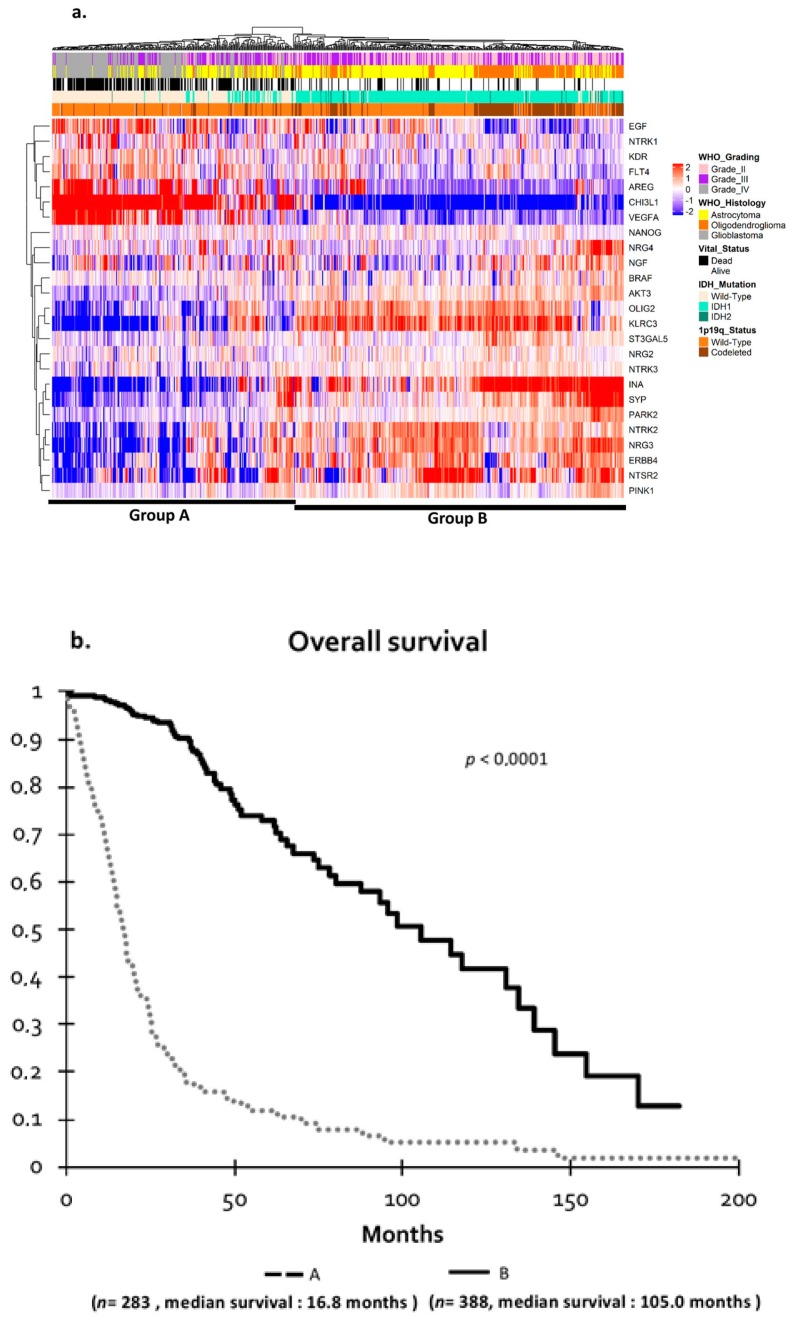
(**a**) Hierarchical clustering of brain tumors based on The Cancer Genome Atlas (TCGA) cohort: Hierarchical clustering of 671 gliomas based on 26 genes. Analysis of TCGA samples was performed as in Figure 1a. (**b**) Overall survival in TCGA cohort by Kaplan–Meier analysis. Survival group A (dotted line) defined unfavorable prognosis, and survival group B (bold line) defined favorable prognosis (*p* = 0.007). *p*-values were calculated using log-rank test.

**Figure 3 cancers-11-00544-f003:**
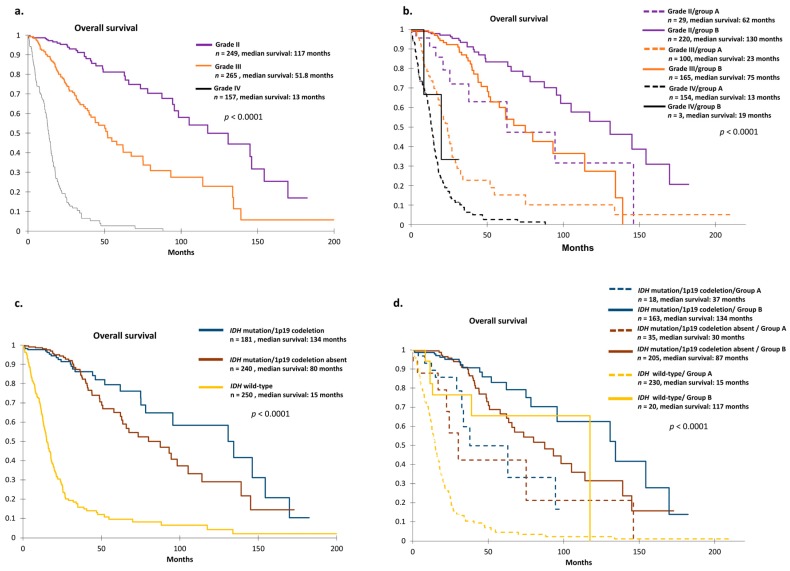
(**a**) Overall survival curves according to grade in TCGA cohort by Kaplan–Meier analysis. *p*-values were calculated using log-rank test. (**b**) Overall survival curves according to grade and 26-gene signature in TCGA cohort. Survival group A (dotted line) defined unfavorable prognosis, and survival group B (solid line) defined favorable prognosis (*p* < 0.0001) by Kaplan–Meier analysis. In subgroups for each grade, the 26-gene signature provided information about prognosis. *p*-values were calculated using log-rank test. (**c**) Kaplan–Meier overall survival curves according to molecular characteristics in TCGA cohort. *p*-values were calculated using log-rank test. (**d**) Overall survival curves according to molecular characteristics and 26-gene signature in TCGA cohort. Survival group A (dotted line) defined unfavorable prognosis, and survival group B (solid line) defined favorable prognosis (*p* < 0.0001). In each subgroup of molecular characteristics, the 26-gene signature further refined prognosis. *p*-values were calculated using log-rank test.

**Figure 4 cancers-11-00544-f004:**
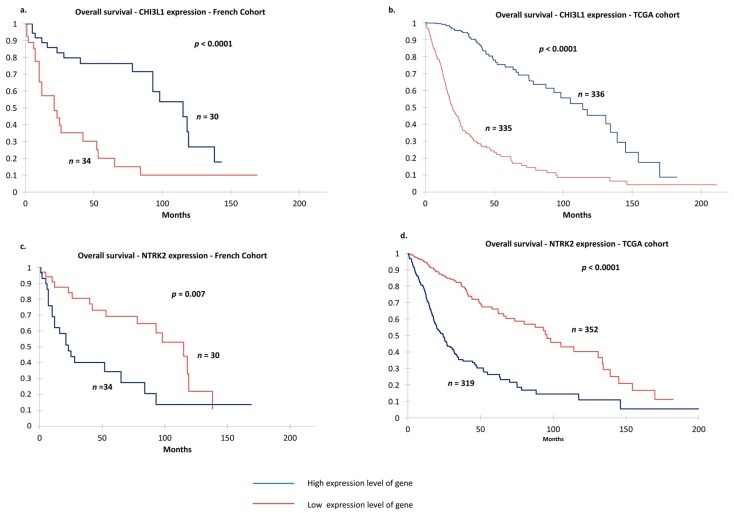
Overall survival according to differential gene expression (median) by Kaplan–Meier analysis. Differential expression of *CHI3L1* in glioma from (**a**) our cohort and (**b**) TCGA cohort. Differential expression of *NTRK2* in glioma from (**c**) our cohort and (**d**) TCGA cohort. Red line represents high gene expression; blue line, low expression. *p*-values were calculated using log-rank test.

**Figure 5 cancers-11-00544-f005:**
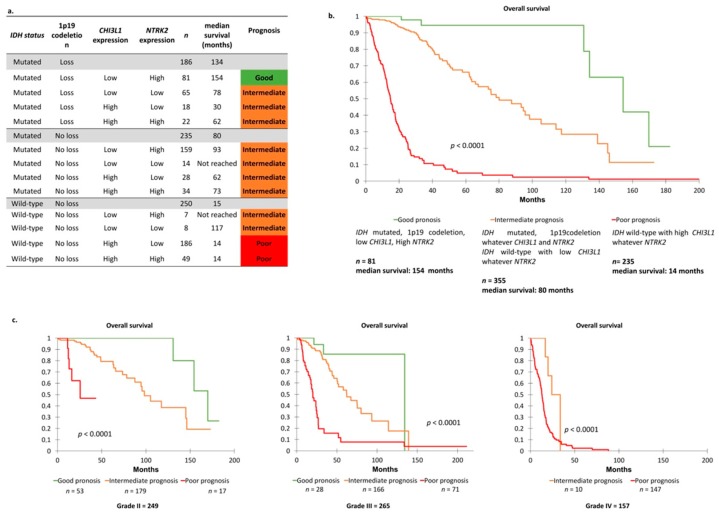
(**a**) Overall survival in glioma molecular groups according *IDH* status, 1p19 co-deletion and *CHI3L1* and *NTRK2* expression. (**b**) Overall survival in three glioma molecular groups by Kaplan–Meier analysis: Three groups with different prognoses were identified (*p* < 0.0001). Three molecular groups were identified: Poor prognosis group combining *IDH* wild-type status, no co-deletion of 1p19q, high *CHI3L1* expression and low *NTRK2* expression; very good prognosis group combining *IDH* mutated, 1p19q co-deletion, low *CHI3L1* expression and high *NTRK2* expression; and intermediate prognosis group combining the other associations (*p* < 0.0001). *p*-values were calculated using log-rank test. (**c**) Overall survival in each grade of glioma according to molecular group by Kaplan–Meier analysis: Three groups with different prognoses were identified (*p* < 0.0001).

**Figure 6 cancers-11-00544-f006:**
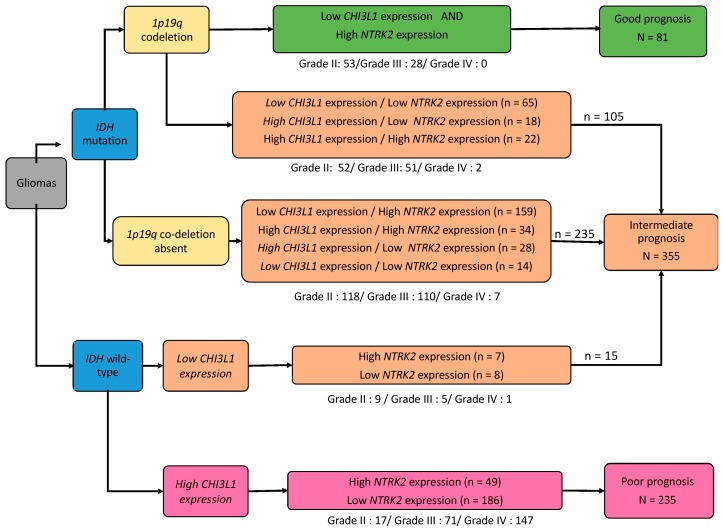
Flowchart summarizing conclusions of the study. Three molecular subgroups based on *IDH* mutation, *1p/19q* co-deletion and *CHI3L1* and *NTRK2* gene expression status were identified and showed distinct clinical presentations with different prognoses.

**Table 1 cancers-11-00544-t001:** Multivariate logistic regression model for *IDH* mutation.

Factors	OR (95% CI)	*p*-Value
Grade (II vs. III vs. **IV**)	0.101 (0.0–0.2)	<0.0001
*CHI3L1* expression (**low** vs. high)	11.8 (6.0–23.2)	<0.0001
*NTRK2* expression (**low** vs. high)	0.3 (0.2–0.5)	0.0001
*1p19* co-deletion (**present** vs. absent)	25.0 (13.7–468.8)	<0.0001

OR, odds ratio; CI, confidence interval.

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
