# Peer review of "CHI3L1, NTRK2, 1p/19q and IDH Status Predicts Prognosis in Glioma"

_cancers, 2019, doi:10.3390/cancers11040544_

Round 1
Reviewer 1 Report
General comments:
The aim of the study is to find biomarkers capable of improving the classification of subgroups of gliomas. This main objective is clearly achieved, obtaining results in terms of patient segregation that are even more striking than those given by traditional biomarkers.
Results showed in the manuscript have a clear value in clinical prognosis.
The manuscript is clearly written although there are numerous misspellings and misprints that should be carefully reviewed. Some of them are pointed below.
line 76 : .“ando analysed whether ithey could refine prognosis according to IDH and 1p19q status in glioma “ . Please, correct the misprints
line 79 :.“chemotherapy-naïve”. Please, correct the misprints.
lines 90-91: Please, rewrite this sentence so it does nor make sense.
line 99: .“prognostic groups, A and (Figure 1a) which were …” I suppose “B” is missed.
line 189: .“molecular molecular groups “. Please, remove molecular.
lines 205-206 .“codeleted tumors based on CHI3L1 and NTRK2 expressions. Median survival in group IDH-mutated, 1p19q co-deleted “. Please, replace codeleted by co-deleted and also along the text.
lines 224-225 .“Since radiotherapy can modify gene expression [21], we only analyzed only the glioma samples resected”. Please, note that “only” is repeated.
Questions:
1) Regarding the possible modification of gene expression after Radiotherapy and/or Chemotherapy, information on the treatment received after the sample was taken is included in Table S1 but not in Table S2 for the TCGA cohort. Do you have this information?
Since this point could be crucial to the reliability of the prediction, I suggest to the authors study if there are differences depending on the previous reception of radiotherapy or not.
If authors dispose of samples collected after Radiotherapy and/or Chemotherapy, they could validate the signature also in samples after treatment which could be reliable in clinic. If it were not possible and this information existed in the TCGA database, it would be interesting to check it at least in this cohort.
2) What do you consider high and low expression? How is this division made? It is not clear to me whether it is done depending on the median or comparing with normal brain tissue.
This should be clearly stated and if it was established by comparing it to normal tissue, the number of normal brain tissue samples included in the pool should be stated.
Authors should indicate this information both when referring to the expression signature of 24 genes and the expression of 2 genes (CHI3L1 and NTRK2).
Author Response
Response to Reviewer 1 comments (Rebuttal letter)
ID: cancers-452453
Reviewer 1:
The aim of the study is to find biomarkers capable of improving the classification of subgroups of gliomas. This main objective is clearly achieved, obtaining results in terms of patient segregation that are even more striking than those given by traditional biomarkers.
Results showed in the manuscript have a clear value in clinical prognosis.
The manuscript is clearly written although there are numerous misspellings and misprints that should be carefully reviewed. Some of them are pointed below.
Point 1:
line 76 : .“ando analysed whether ithey could refine prognosis according to IDH and 1p19q status in glioma “ . Please, correct the misprints
Line 80 : We corrected the misprints
Line 79 :.“chemotherapy-naïve”. Please, correct the misprints.
Line 83: We corrected the misprints
lines 90-91: Please, rewrite this sentence so it does nor make sense.
Line 94-96 : we replaced the confusing sentence: “ Univariate analysis detected that the expression of 29 genes of interest (p < 0.05) (Figure S1, left) and multivariate analysis showed 30 genes (Figure S1, right). The two aforementioned gene lists thus generated a shared set of 26 relevant genes” by “Univariate analysis detected 29 genes of interest (p < 0.05) (Supplementary Figure S1, left), whereas principal component analysis detected 30 genes of interest (Supplementary Figure S1, right). By combining these two techniques, we identified a relevant 26-gene panel»
line 99: .“prognostic groups, A and (Figure 1a) which were …” I suppose “B” is missed.
Line 104: We corrected the misprints
line 189: .“molecular molecular groups “. Please, remove molecular.
Line 194 : we removed molecular
lines 205-206 .“codeleted tumors based on CHI3L1 and NTRK2 expressions. Median survival in group IDH-mutated, 1p19q co-deleted “. Please, replace codeleted by co-deleted and also along the text.
Line 211, 212, 213 we replace codeleted by co-deleted
lines 224-225 .“Since radiotherapy can modify gene expression [21], we only analyzed only the glioma samples resected”. Please, note that “only” is repeated.
Line 235-236 : we removed a “only”
Point 2:
Regarding the possible modification of gene expression after Radiotherapy and/or Chemotherapy, information on the treatment received after the sample was taken is included in Table S1 but not in Table S2 for the TCGA cohort. Do you have this information?
The reviewer’s suggestion is interesting. In the TCGA cohort, we found a negligible number of patients treated with radiotherapy (only 4 out of 671, 3 grade III and 1 grade II), the name is not very suggestive: "history ionizing radiotherapy to head". Unfortunately, no precise information was available on when these patients were exposed to ionizing radiation (before or after biopsy) and which tumor was treated. Under these circumstances, we preferred not to include these data in Table S2.
Since this point could be crucial to the reliability of the prediction, I suggest to the authors study if there are differences depending on the previous reception of radiotherapy or not.
If authors dispose of samples collected after Radiotherapy and/or Chemotherapy, they could validate the signature also in samples after treatment which could be reliable in clinic. If it were not possible and this information existed in the TCGA database, it would be interesting to check it at least in this cohort.
In our French cohort, patients were never biopsied after radiotherapy.
In the TCGA cohort, only 4 patients on 671 have been treated by radiotherapy. This number of radiotherapy-treated patients was neglecting compared to the whole population and unfortunately, no information regarding the ionizing radiation timing were available. Thus, it would not be appropriate to analyze such a small subpopulation for which only imprecise data are available.
Point 3:
What do you consider high and low expression? How is this division made? It is not clear to me whether it is done depending on the median or comparing with normal brain tissue.
This should be clearly stated and if it was established by comparing it to normal tissue, the number of normal brain tissue samples included in the pool should be stated. Authors should indicate this information both when referring to the expression signature of 24 genes and the expression of 2 genes (CHI3L1 and NTRK2).
To define high and low expression, we used the median of each gene. As suggested in reviewer’s comment, we added this sentence Line 387-388 “To distinguish low and high expression of a gene, the median cutoff was chosen as previously described in the literature [56]” to support our methodology.
Line 179, we added NTRK2 (median: 14.76) and CHI3L1 (median: 8.98) and in the caption of figure 4 (line 184), we added median “Overall survival according to gene differential expression (median) by Kaplan Meier analysis ».
Normal brain tissues (2 samples) were only used to normalize TLDA analysis and was not used to define high and low expression. We thanks to reviewer for his suggestion and we added that two samples of brain tissues were used (Line 334).
Figure 2 : we removed this confusing sentence : “fold change values (as compared to normal brain) was generated using Pearson correlation distance measure and average linkage”

Reviewer 2 Report
In this study titled “CHI3L1, NTRK2, 1p/19q and IDH status predicts prognosis in glioma”, Elise and colleagues investigated the gene expression profile of 96 selected genes in 64 cases of newly diagnosed diffuse glioma and identified a 26-gene panel which can stratify gliomas into two distinct prognostic groups. The prognostic value of the 26-gene panel was validated by the TCGA dataset. Furthermore, they also identified CHI3L1 and NTRK2 as the two most differentially expressed genes among the 26-gene panel and each of the genes show prognostic value in gliomas. By supplementing the two genes to IDH mutation and 1p/19q codeletion, the authors were able to stratify diffuse gliomas into three prognostic groups which refined the prognostic stratification based on IDH mutation and 1p/19q codeletion. Overall the study was interesting and provided data which may have potential clinical implications.
1. It was unknown what was the selection criteria for those “96 selected genes” to be included in the gene expression profile. Was it based on the authors’ previous studies or any pathway analysis which they identified these 96 genes as important genes in glioma? Some background of this 96-gene panel should be included in the introduction.
2. The authors only perform univariate analysis to examine the prognostic values of the 26-gene set, CHI3L1 expression and NTRK2 expression. No multivariable analysis was done to examine whether they are independent prognostic marker in gliomas. Clinical parameters (e.g. age, total resection), pathological parameters (e.g. histologic grade, histologic phenotype) and molecular parameters (e.g. IDH, 1p/19q) with well-known prognostic impact should be included in the multivariable analysis. All these data available in the author’s original cohort as well as the TCGA dataset.
3. Was there any association between histologic grade and CHI3L1 expression / NTRK2 expression? Can the prognostic scheme in figure 5 be applied independently in grade II, grade III and grade IV gliomas?
4. In the Discussion, second paragraph, the author said that “However, only a few studies evaluated the impact of a gene signature across low and high grade tumors” (Page 8, line 230 – 231). There are indeed numerous genomics studies which include expression profiling covering grades II to IV diffuse gliomas. The authors may want to revise this statement.
5. Discussion, page 9, line 272 – 273, “CHI3L1 expression and IDH status might depend on epigenetic mechanisms such as DNA methylation remodelling [42].” It’s unknown why the authors put down this statement. The reference they cited was a study comparing the molecular differences between MGMT methylated IDH wildtype glioblastomas and MGMT unmethylated IDH wildtype glioblastomas. There was no data about CHI3L1 in this study. Also previous studies have demonstrated that IDH mutation lead to genome hypermethylation in diffuse gliomas. The reviewer considers this statement to be erroneous and should be deleted.
6. Discuss, page 9, line 286, “Based on the data presented here, we suggest that CHI3L1 and NTRK2 expression define molecular subgroups and might determine prognosis in glioma.” The data presented here is insufficient to conclude the two genes as molecular subgroup defining markers. The two genes only serve as potential surrogate marker to identify certain subgroups with distinct prognosis.
7. The authors should expand the discussion about the potential use of the markers in clinical setting, limitations as well as potential technical difficulties.
8. The authors should carefully proofread the manuscript and correct the typos. Here are some examples.
a. Page 2, line 76 – “ando analysed whether ithey…”
b. Page 3, line 145 – “only NGF lost it prognostic significance….”
c. Page 6, line 188-189 – “Then were identified three molecular molecular groups were identified …….”
d. Page 7, line 211-212 – “demonstrated that low CHI3L1, high NTRK2, grade expressions and 1p19q codeletion seemed to be independently predictors…”
Author Response
Response to Reviewer 2 comments (Rebuttal letter)
ID: cancers-452453
Reviewer 2:
In this study titled “CHI3L1, NTRK2, 1p/19q and IDH status predicts prognosis in glioma”, Elise and colleagues investigated the gene expression profile of 96 selected genes in 64 cases of newly diagnosed diffuse glioma and identified a 26-gene panel which can stratify gliomas into two distinct prognostic groups. The prognostic value of the 26-gene panel was validated by the TCGA dataset. Furthermore, they also identified CHI3L1 and NTRK2 as the two most differentially expressed genes among the 26-gene panel and each of the genes show prognostic value in gliomas. By supplementing the two genes to IDH mutation and 1p/19q codeletion, the authors were able to stratify diffuse gliomas into three prognostic groups which refined the prognostic stratification based on IDH mutation and 1p/19q codeletion. Overall the study was interesting and provided data which may have potential clinical implications.
Point 1:
It was unknown what was the selection criteria for those “96 selected genes” to be included in the gene expression profile. Was it based on the authors’ previous studies or any pathway analysis which they identified these 96 genes as important genes in glioma? Some background of this 96-gene panel should be included in the introduction.
As suggested in reviewer’s comment we modified introduction
Line 64-65, we added publications of our laboratory to highlight that our 96 selected genes were based on previous studies. We added one sentence: “So, in this study we analyzed 96 genes considered as relevant based on our previous work [14]”.
Line 70-78: “Our previous study allowed us to present evidence that eight glycosylation-related genes were upregulated in the most aggressive and undifferentiated glioblastoma cells [14]. In parallel, the family of tyrosine receptor kinases (TRKs) and p75NTR are already known to be involved in tumor cell survival. For instance, TrkB and TrkC receptors promote the growth of brain tumor–initiating cells, and p75NTR promotes glioma invasion [15–17]. Finally, we recently showed that some autophagic factors and neurotrophin pathways cooperate to contribute to tumor cell aggressiveness [18]. Thus, genes encoding Trks and glycosylation-related genes could be of interest to complement the current IDH mutation¬based classification.”
Point 2:
The authors only perform univariate analysis to examine the prognostic values of the 26-gene set, CHI3L1 expression and NTRK2 expression. No multivariable analysis was done to examine whether they are independent prognostic marker in gliomas. Clinical parameters (e.g. age, total resection), pathological parameters (e.g. histologic grade, histologic phenotype) and molecular parameters (e.g. IDH, 1p/19q) with well-known prognostic impact should be included in the multivariable analysis. All these data available in the author’s original cohort as well as the TCGA dataset.
We agree with this suggestion.
To answer to reviewer’s suggestion, we added a supplementary Table S3 (line 416, Table S3 description) in which we performed univariate versus multivariate analysis to examine the prognostic value of our genes. Considering data from TCGA, the following sentence was added Line 223-225: “Multivariate Cox model analysis revealed independent markers of prognosis: age of diagnosis (hazard ratio (HR): 1.039; 95% confidence interval (CI): 1.028–1.051; p < 0.0001), grade of glioma (p < 0.05), and molecular combined group (p < 0.05). (Supplementary Table S3)”. We clearly noted in discussion that it was possible to define patient groups based on IDH status and 1p19q co-deletion status as described by Eckel et al. in order to achieve multivariate analysis and determine the prognostic value of novel molecular groups. Unfortunately, this methodology could not be applied on our French cohort composed of only 64 patients. Indeed, the size of each group was too limited and statistical analysis was not achievable.
Point 3:
Was there any association between histologic grade and CHI3L1 expression / NTRK2 expression? Can the prognostic scheme in figure 5 be applied independently in grade II, grade III and grade IV gliomas?
Histologic grade and CHI3L1 expression / NTRK2 expression were independent factors as it was shown in multivariate analysis (Table 1). To highlight this point, we added as suggested in reviewer’s comment in figure 5c, prognostic scheme in each grade of glioma and we added a new legend.
We added this sentence in line 199-200: “Prognostic populations were found in each grade of glioma suggesting that the prognostic scheme could be applied independently in grade II, grade III and grade IV gliomas” (Figure 5c).
Point 4:
In the Discussion, second paragraph, the author said that “However, only a few studies evaluated the impact of a gene signature across low and high grade tumors” (Page 8, line 230 – 231). There are indeed numerous genomics studies which include expression profiling covering grades II to IV diffuse gliomas. The authors may want to revise this statement.
We thank the referee for this remark
We replaced theses sentences “However, only a few studies evaluated the impact of a gene signature across low and high grade tumors. These latter studies showed that gene expression profiling provides additional information to distinguish between glioma subtypes [22–24]” by “ Moreover, some studies showed that gene expression profiling provides additional informations to discriminate between glioma subtypes from grade II to IV.” in line 241-243.
Point 5:
Discussion, page 9, line 272 – 273, “CHI3L1 expression and IDH status might depend on epigenetic mechanisms such as DNA methylation remodelling [42].” It’s unknown why the authors put down this statement. The reference they cited was a study comparing the molecular differences between MGMT methylated IDH wildtype glioblastomas and MGMT unmethylated IDH wildtype glioblastomas. There was no data about CHI3L1 in this study. Also previous studies have demonstrated that IDH mutation lead to genome hypermethylation in diffuse gliomas. The reviewer considers this statement to be erroneous and should be deleted.
We thank the reviewer for this remark and removed this sentence and reference in the revised manuscript.
Point 6:
Discuss, page 9, line 286, “Based on the data presented here, we suggest that CHI3L1 and NTRK2 expression define molecular subgroups and might determine prognosis in glioma.” The data presented here is insufficient to conclude the two genes as molecular subgroup defining markers. The two genes only serve as potential surrogate marker to identify certain subgroups with distinct prognosis.
We agree and replaced the previous sentence: “Based on the data presented here, we suggest that CHI3L1 and NTRK2 expression define molecular subgroups and might determine prognosis in glioma.” by “Based on the data presented here, we suggest that CHI3L1 and NTRK2 were two potential surrogate markers to identify certain subgroups with distinct prognosis” in line 295-296.
Point 7:
The authors should expand the discussion about the potential use of the markers in clinical setting, limitations as well as potential technical difficulties.
We thank the reviewer for this remark and develop it in discussion
Line 302-311 “In the future, better treatment outcomes for patients with a poor prognosis (IDH wild-type, absence of 1p19 co-deletion, high CHI3L1 expression, and low NTRK2 expression) might be proposed by combining conventional chemotherapy (radiotherapy and temozolomide) with CHI3L1-targeted inhibitors, especially in grade IV [46]. In grade II patients with a very good prognosis, therapeutic de-escalation or treatment with NTRK2 inhibitor may be considered.
These results will have to be validated in a prospective investigation in order to develop a robust protocol for microarray analysis, to test the interobserver reproducibility of our approach, and to confirm the prognostic value of these biomarkers and/or their potential as therapeutic targets. CHI3L1 and NTRK2 status could be investigated in tumors by using frozen tissues, or in the future by using formalin-fixed paraffin-embedded sections to facilitate their use in routine practice [47].”
Point 8:
The authors should carefully proofread the manuscript and correct the typos. Here are some examples.
a. Page 2, line 76 – “ando analysed whether ithey…”
New Line 80 : We corrected the misprints
b. Page 3, line 145 – “only NGF lost it prognostic significance….”
New line 150 : We corrected the misprints
c. Page 6, line 188-189 – “Then were identified three molecular molecular groups were identified …….”
Line 193 : we removed molecular
d. Page 7, line 211-212 – “demonstrated that low CHI3L1, high NTRK2, grade expressions and 1p19q codeletion seemed to be independently predictors…”
New Line 219 : We corrected the misprints

Round 2
Reviewer 2 Report
The reviewer's comments and concerns are sufficiently addressed.